# Sulforaphane Is Protective against Warm Ischemia/Reperfusion Injury and Partial Hepatectomy in Rats

**DOI:** 10.3390/ijms25010579

**Published:** 2024-01-01

**Authors:** Richi Nakatake, Tetsuya Okuyama, Yuki Hashimoto, Morihiko Ishizaki, Hidesuke Yanagida, Hiroaki Kitade, Katsuhiko Yoshizawa, Mikio Nishizawa, Mitsugu Sekimoto

**Affiliations:** 1Department of Surgery, Kansai Medical University, Hirakata 573-1010, Osaka, Japan; okuyamat@hirakata.kmu.ac.jp (T.O.); hashimoy@hirakata.kmu.ac.jp (Y.H.);; 2Department of Innovative Food Sciences, School of Food Sciences and Nutrition, Mukogawa Women’s University, 6-46 Ikebiraki-cho, Nishinomiya 663-8558, Hyogo, Japan; yoshizak@mukogawa-u.ac.jp; 3Department of Biomedical Sciences, College of Life Sciences, Ritsumeikan University, 1-1-1 Nojihigashi, Kusatsu 525-8577, Shiga, Japan

**Keywords:** sulforaphane, ischemia/reperfusion injury, partial hepatectomy, primary cultured hepatocytes, nuclear factor-kappa B

## Abstract

Sulforaphane (SFN) has various beneficial effects on organ metabolism. However, whether SFN affects inflammatory mediators induced by warm hepatic ischemia/reperfusion injury (HIRI) is unclear. To investigate the hepatoprotective effects of SFN using an in vivo model of HIRI and partial hepatectomy (HIRI + PH), rats were subjected to 15 min of hepatic ischemia with blood inflow occlusion, followed by 70% hepatectomy and release of the inflow occlusion. SFN (5 mg/kg) or saline was randomly injected intraperitoneally 1 and 24 h before ischemia. Alternatively, ischemia was prolonged for 30 min to evaluate the effect on mortality. The influence of SFN on the associated signaling pathways was analyzed using the interleukin 1β (IL-1β)-treated primary cultured rat hepatocytes. In the HIRI + PH-treated rats, SFN reduced serum liver enzyme activities and the frequency of pathological liver injury, such as apoptosis and neutrophil infiltration. SFN suppressed tumor necrosis factor-alpha (TNF-α) mRNA expression and inhibited nuclear factor-kappa B (NF-κB) activation by HIRI + PH. Mortality was significantly reduced by SFN. In IL-1β-treated hepatocytes, SFN suppressed the expression of inflammatory cytokines and NF-κB activation. Taken together, SFN may have hepatoprotective effects in HIRI + PH in part by inhibiting the induction of inflammatory mediators, such as TNF-α, via the suppression of NF-κB in hepatocytes.

## 1. Introduction

Hepatic ischemia/reperfusion injury (HIRI) is common in a variety of clinical situations, including hepatobiliary surgery and liver transplantation. The Pringle maneuver (PM) is a non-selective inflow occlusion technique that is widely used for blood loss control during continuous or intermittent portal triad clamping [1,2]. Warm HIRI occurs when hepatic inflow is transiently clamped and is a major risk factor for postoperative complications in the setting of massive hepatic resection [3]. The main molecular mechanisms of HIRI, including apoptosis, oxidative stress, neutrophil infiltration, and release of inflammatory cytokines, remain to be elucidated [4,5]. When HIRI is induced, the nuclear factor-kappa B (NF-κB) pathway is activated, leading to the development of inflammation and exacerbation of liver tissue damage [6,7]. NF-κB is an important transcription factor that regulates the synthesis of various cytokines and chemokines in immunologic stimuli [8,9]. In the HIRI process, apoptosis of hepatocytes is the core of liver damage caused by various pathways, including the NF-κB pathway [10]. Appropriate control of inflammatory responses during the perioperative period is crucial to prevent organ damage.

In recent years, interest in the nutritional and medicinal value of sulforaphane (SFN; Figure 1A) has grown. SFN, which has been identified in broccoli [11], is one of the most fascinating phytochemicals known because it protects cells under aerobic conditions from toxic carcinogens, electrophiles, and oxidants that damage DNA through the induction of the network of phase 2 detoxifying and antioxidant enzymes and the suppression of inflammatory responses [12,13]. The principal cytoprotective properties of SFN are mediated by upregulation of the transcription factor NF-E2-related factor 2 (Nrf2) and by anti-inflammatory mechanisms, such as the inhibition of the NF-κB pathway [14,15]. However, there are few reports on the efficacy of SFN in improving hepatic abnormalities in humans to date. Many animal studies have been performed to demonstrate the hepatoprotective effects of SFN against hepatotoxic chemicals, such as cisplatin [16], microcystin [17], carbon tetrachloride [18], acetaminophen [19], and D-galactosamine and lipopolysaccharide (LPS) combination [20]. Furthermore, SFN has been suggested to be potent in preventing lifestyle diseases caused by excess alcohol consumption [21] and high-energy diets [22,23]. These hepatoprotective effects have been suggested to occur because of the superior induction capacity of SFN for phase II cytoprotective proteins, such as detoxification and antioxidant enzymes, via the activation of the transcription factor—Nrf2. Regarding the anti-inflammatory properties of SFN, SFN inhibits the expression of inflammatory factors (tumor necrosis factor-alpha, TNF-α; interleukin 1β, IL-1β; and IL-6) and inflammatory mediator enzymes (inducible nitric oxide synthase and cyclooxygenase 2) and the activation of NF-κB in LPS-treated mouse RAW 264.7 macrophages [24] and goat mammary epithelial cells [25]. The inhibition of TNF-α and inflammatory mediators (nitric oxide and prostaglandin E_2_) production has been proposed as a useful approach for the treatment of various inflammatory diseases and a potential chemoprevention strategy [25,26].

In this study, the hepatoprotective effects of SFN were examined using two models: an in vivo warm HIRI and a partial hepatectomy model (HIRI + PH) and an in vitro model of liver injury using primary rat hepatocyte cultures.

## 2. Results

### 2.1. SFN Decreases Pathological Changes in Liver Injury

The livers of HIRI + PH-treated rats were examined histologically after 3 h and 6 h (Figure 1B). Figure 2A–E present representative images of the hematoxylin and eosin (H&E)-stained sections. Liver injury was characterized by hemorrhagic manifestations, focal necrosis, and inflammatory cell infiltration (Figure 2B,D) compared to the control (Figure 2A). Ballooning degeneration of the hepatocytes was also observed. SFN reduced the frequency and extent of these pathological changes (Figure 2C,E). Damage was graded using the Suzuki score in Figure 2F; SFN significantly reduced the grade of liver injury by 0.25-fold at 3 h (*p* = 0.044) and by 0.38-fold at 6 h (*p* < 0.001), respectively, compared with HIRI + PH-only treatment.

In Figure 3A–E, representative images of liver sections stained by terminal deoxynucleotidyl transferase-mediated deoxyuridine nick-end labeling (TUNEL) are shown. HIRI + PH increased apoptosis in the hepatocytes at 6 h (Figure 3D), whereas SFN markedly decreased apoptosis by 0.39-fold compared to HIRI + PH-only treatment (*p* = 0.002; Figure 3E,F).

Next, the effects of HIRI + PH treatment on neutrophil infiltration in the liver were evaluated by immunostaining for myeloperoxidase (MPO) (Figure 4). HIRI + PH markedly increased neutrophil infiltration in the liver at 6 h (Figure 4A,B,D), whereas SFN markedly decreased neutrophil infiltration (by 0.40-fold, *p* = 0.002; Figure 4F). A summary of the results is provided in Appendix A.

### 2.2. SFN Attenuates Cell Damage in the Livers of HIRI + PH-Treated Rats

HIRI + PH resulted in a rapid increase in aspartate aminotransferase (AST) and alanine aminotransferase (ALT) serum levels after reperfusion. In contrast, AST (6 h) and ALT (3 h) serum levels were reduced by 0.47-fold (*p* = 0.018) and 0.24-fold (*p* = 0.007), respectively, by treatment with SFN compared to HIRI + PH-only treatment (Figure 5).

### 2.3. SFN Affects Inflammation-, Apoptosis-, and Liver Generation-Related Gene Expression in HIRI + PH-Treated Rats

The HIRI + PH and SFN groups had significantly lower mRNA expression of TNF-α (3 h, 0.52-fold, *p* = 0.012, Figure 6A) and C-X-C motif chemokine ligand 1 (CXCL1; 6 h, 0.27-fold, *p* = 0.007, Figure 6B) genes than the expression in the HIRI + PH group. The HIRI + PH and SFN group had significantly higher mRNA expression in the IL-10 gene than the expression in the HIRI + PH group (6 h, 2.21-fold, *p* = 0.013, Figure 6C). Moreover, mRNA expression of myeloid cell leukemia 1 (MCL1), a member of the Bcl-2 family, was significantly enhanced by SFN (3 h, 1.84-fold, *p* = 0.032; 6 h, 1.87-fold, *p* = 0.005, Figure 6D). SFN significantly increased hepatocyte growth factor (HGF) mRNA expression at 6 h (2.67-fold, *p* < 0.001, Figure 6E).

### 2.4. SFN Decreases NF-κB Activation in HIRI + PH-Treated Rats

NF-κB activation is an upstream factor of inflammatory cytokine transcription. The electrophoretic mobility shift assay (EMSA) using the nuclear extract of the livers indicated that SFN suppressed the activation of NF-κB by HIRI + PH by 0.73-fold at 6 h with a *p*-value of 0.024 (Figure 7).

### 2.5. SFN Decreases Mortality in Prolonged HIRI + PH-Treated Rats

The protective effect of SFN in a lethal model with prolonged ischemic duration and PH was investigated (Figure 1C). The survival rate was only 0.524 after HIRI + PH treatment. However, the administration of SFN significantly reduced long-term ischemia-induced mortality, with a survival rate of 0.927 (Figure 8; *p* = 0.0402).

### 2.6. SFN Inhibits Inflammation-Related Gene Expression and Blocks NF-κB Activation in Cultured Hepatocytes

We further assessed the underlying mechanisms by which SFN suppresses the inflammatory gene transcription using rat primary cultured hepatocytes [27,28]. First, we found that SFN dose–dependently suppressed nitric oxide production (Appendix A) without showing cytotoxicity up to 10 μM in IL-1β-stimulated hepatocytes. The half-maximal inhibitory concentration (IC_50_) value was calculated to be 1.20 μM. Since the suppression of nitric oxide production is closely related to the suppression of inflammatory gene mRNA expression, not only by a variety of drugs but also by several natural phytochemicals [29,30,31,32], SFN was used at 10 μM for further analysis. SFN decreased TNF-α (0.02–0.05-fold, *p* < 0.001), IL-1β (0.06–0.19-fold, *p* < 0.001), IL-6 (0.07–0.32-fold, *p* < 0.001), and CXCL1 mRNA expression (0.29–0.65-fold, *p* < 0.005) in IL-1β-treated rat-cultured hepatocytes (Figure 9A–D). IL-1β stimulated NF-κB activation [32], whereas SFN inhibited NF-κB activation (Figure 9E).

## 3. Discussion

In this study, the hepatoprotective effects of SFN were investigated in the in vivo HIRI + PH model and the in vitro liver injury model. The results suggested several possible mechanisms for the effects. In this HIRI + PH rat model, SFN reduced the frequency of pathological liver injuries (Figure 2), including apoptosis (Figure 3) and the infiltration of neutrophils (Figure 4), and attenuated changes in serum levels of AST and ALT (Figure 5). TNF-α contributes to neutrophil recruitment in the liver, which is ultimately involved in the later phases of injury [33]. SFN significantly decreased the levels of TNF-α and CXCL1 mRNA and increased the levels of IL-10 mRNA in the livers of HIRI + PH-treated rats (Figure 6A–C). SFN treatment upregulated MCL1 mRNA (Figure 6D), a negative regulator of apoptosis, thereby suppressing HIRI + PH-induced apoptosis. NF-κB functions as an important regulator of hepatocyte regeneration and anti-apoptosis [33]. In the EMSA of nuclear extracts from the liver, SFN inhibited the activation of NF-κB by HIRI + PH (Figure 7). The cytoprotective and anti-inflammatory effects of SFN may be partially attributed to its ability to modulate the transcription factor NF-κB. Moreover, the administration of SFN improved survival in a more severe HIRI model of 30 min of ischemia followed by 70% hepatectomy (Figure 8). SFN may improve survival rates by suppressing the expression of the inflammatory cytokines and chemokines and enhancing the expression of the anti-inflammatory cytokine and anti-apoptotic genes.

The two main types of liver cells affected by HIRI are hepatocytes and sinusoidal endothelial cells, the cells differ in their sensitivity to different types of warm or cold ischemia, with hepatocytes being more sensitive to warm ischemia [34]. Hepatocyte apoptosis is a feature of liver inflammation and is associated with the production of a variety of inflammatory cytokines. The suppression of the inflammatory response with SFN administration may contribute to the marked reduction in hepatocyte apoptosis. After identifying the efficacy of SFN in the near-clinical setting of hepatic ischemia/reperfusion and partial hepatectomy, we next confirmed the mechanisms using primary cultured hepatocytes. In rat models of acute liver injury induced by ischemia/reperfusion or partial hepatectomy and in primary rat hepatocytes stimulated by IL-1β, several drugs and bioactive compounds with hepatoprotective effects have been shown to decrease the production of inflammatory mediators, such as TNF-α, through the suppression of NF-κB [35,36,37,38]. TNF-α increases the release of other molecules, such as IL-6 [39] and numerous chemokines (including CXCL1) [40]. SFN reportedly interacts with the NF-κB subunits, directly reducing their ability to bind DNA [24]. SFN decreased levels of inflammatory cytokines and chemokine mRNA (Figure 9A–D), the end products of NF-κB, indicating that the hepatoprotective effects were achieved through the inhibition of the expression of these inflammatory genes. As with in vivo experiments, SFN also suppressed NF-κB activation in cultured hepatocytes (Figure 9E). Several intracellular signaling pathways are known to be involved in NF-κB activation in primary cultured rat hepatocytes, including IκBα and Akt upon stimulation with IL-1β. However, SFN did not affect the induction of IκBα degradation or Akt activation. It remains to be elucidated whether SFN suppresses NF-κB activity through other signaling pathways or upstream factors.

Promoting liver regeneration may prevent liver failure after partial hepatectomy [41]. In rats, the process of liver regeneration in rats begins with an increase in the levels of various signaling molecules 3 h after hepatectomy, peaks at two to three days, and ends with the enlargement of the remnant lobe to the original liver size at five to seven days [42]. The prime phase of liver regeneration occurs 24 h after partial hepatectomy, as assessed by the liver-to-body weight ratio and immunohistochemical markers of proliferative events, such as Ki-67 [43]. The hepatoprotective effects of SFN may involve liver regeneration after HIRI + PH treatment. SFN significantly enhanced HGF expression at 6 h of HIRI + PH treatment (Figure 6E), suggesting that SFN may affect liver regeneration via HGF. The activation of the Nrf2 transcription factor, an intracellular target of SFN, has been reported to promote liver regeneration [44]. These results suggest that SFN may promote liver regeneration, even in HIRI + PH; whether SFN promotes liver regeneration through these factors or other mechanisms should be analyzed after assessing liver regeneration by the liver-to-body weight ratio and immunohistochemical markers.

These findings suggest that the preoperative administration of SFN may be a potential therapeutic option to prevent liver injury after hepatic resection in humans. Although the dose of SFN used in this study (5 mg/kg) is above the dietary dose, this dose is similar to a dose (3 mg/kg) that has been used in previous studies using rat models [20]. Adverse effects were observed with SFN administration at 25 mg/kg. However, the dosages utilized in the in vivo and the in vitro experiments in this study are relatively higher than those used in the human diet [45], thus serving as a limitation in these models.

## 4. Materials and Methods

### 4.1. Materials

SFN (molecular weight, 177.29) was purchased from Cayman Chemical (≥99.0%, Ann Arbor, MI, USA) and LKT Laboratories, Inc. (≥99.0%, St. Paul, MN, USA). Recombinant human IL-1β (2 × 10^7^ U/mg protein) was obtained from MyBioSource, Inc. (San Diego, CA, USA). Male Sprague–Dawley (240–260 g; 6 weeks old) and Wistar rats (180–220 g; 5 weeks old) were obtained from the Jackson Laboratory, Inc. (Yokohama, Japan). Sprague–Dawley rats were used in the previous in vivo rat model of HIRI [35]. On the other hand, the in vitro model of liver injury has been established using the hepatocytes prepared from Wister rat livers [36,46,47,48,49]. In this study, the most appropriate strain was applied to each model. The rats were housed at 22 °C with a 12 h light/dark cycle for ≥7 days to allow acclimatization before experiments. Food and water were provided ad libitum. Animal care and experiments were performed in accordance with the standards outlined in the ARRIVE [50] and PREPARE [51] guidelines. The study protocol was approved by the Animal Care Committee of Kansai Medical University (Osaka, Japan) (approval No. 22-039 and No. 22-040).

### 4.2. Induction of Hepatic Warm Ischemia/Reperfusion Injury and Partial Hepatectomy in Rats

For the in vivo HIRI + PH, male Sprague–Dawley rats (272.5 ± 29.17 g; 7 weeks old after 7-day acclimatization) were anesthetized with 3% isoflurane during 15 min PM by clamping the hepatoduodenal ligament. After 70% hepatectomy, the inflow occlusion was released by declamping. Liver and blood samples were collected from rats after 3 and 6 h (Figure 1B) or from untreated rats as a naïve control. In the survival experiment, the hepatoduodenal ligament was clamped for 30 min before PH. After 70% PH, the remnant liver was reperfused, and all rats were observed for 7 days to determine their survival (Figure 1C) [52]. The rats were randomly assigned to receive intraperitoneal injections of either SFN (5 mg/kg) or saline 1 and 24 h before PM. The route and dose of SFN administration were modified from a previous in vivo study [20]. We followed the NIH Office of Animal Care and Use [53] score and severity assessment to assess the animals following liver resection [54]. The rats were euthanized when they appeared weak and moribund due to the progression of liver failure, congestion, and multi-organ failure.

### 4.3. Culture of Primary Rat Hepatocytes

Perfusion of collagenase (FUJIFILM Wako Pure Chemical Corp., Osaka, Japan) was used to isolate hepatocytes from male Wistar rats (200–250 g; 6–7 weeks old) [47,55]. The isolated hepatocytes were suspended in Williams’ medium E at 6 × 10^5^ cells/mL, seeded into 35 mm plastic dishes (2 mL/dish; Falcon Plastics, Oxnard, CA, USA), and incubated at 37 °C in a humidified chamber (5% CO_2_). The medium contained 10% fetal calf serum, N-(2-hydroxyethyl)piperazine-N′-2-ethanesulfonic acid (5 mmol/L), penicillin (100 U/mL), streptomycin (100 μg/mL), amphotericin B (0.25 μg/mL), aprotinin (0.1 μg/mL; Roche, Basel, Switzerland), dexamethasone (10 nmol/L), and insulin (10 nmol/L). After 2 h, the medium was replaced with a fresh serum-free medium (1.5 mL/dish). After another 3 h, the medium was again replaced with a fresh serum- and hormone-free medium (1.5 mL/dish), and the cells were cultured overnight before use. SFN was first dissolved in dimethylsulfoxide at 10 mM. On the day after cell culture, the hepatocytes were washed with a fresh serum- and hormone-free medium and incubated with IL-1β (1 nmol/L). SFN was diluted 1000-fold in the same medium (final concentration, 10 μM).

### 4.4. Histopathological Analysis of the In Vivo Rat Model

The excised liver specimens were fixed in 4% paraformaldehyde in a phosphate buffer and embedded in paraffin. Three to four animals per group and three to five sites per animal were used for slide preparation. Sections of 3–5 μm in thickness were cut. After staining with H&E, the severity of lesions was graded on a 4-point scale (0 = no change, 1 = minimal, 2 = mild, 3 = moderate, and 4 = marked) for sinusoidal congestion, cytoplasmic vacuolation of hepatocytes, and parenchymal necrosis, as described by Suzuki et al. [56]. The toxicologic pathologist (K.Y.) certified by the International Federation of Societies of Toxicologic Pathologists performed histopathological evaluations according to previously defined histopathological terminology and diagnostic criteria. Apoptotic bodies in the hepatocyte nuclei were detected by TUNEL staining using an in situ Apoptosis Detection Kit (MK500; Takara Bio Inc., Kusatsu, Shiga, Japan). Neutrophil infiltration was evaluated by immunostaining with MPO using rabbit polyclonal anti-MPO antibodies (Dako, Carpinteria, CA, USA) before counterstaining with hematoxylin. The number of TUNEL- and MPO-positive cells per square millimeter was counted by analysts blinded to the treatment.

### 4.5. Serum Biochemical Analysis of the In Vivo Rat Model

Serum AST and ALT levels were quantified using a Transaminase CII Test WAKO (code 431-30901, FUJIFILM Wako Pure Chemical Corp., Osaka, Japan, range 3.9~500 Karman units for AST and 2.4~500 Karman units for ALT, sensitivity OD_555_ = 0.240~0.320 for 100 Karman units of AST and OD_555_ = 0.290~0.370 for 100 Karman units of ALT, inter-assay ≤ ±30% for AST and ALT, and intra-assay CV ≤ 3% for AST and ≤8% for ALT). The sera were diluted by distilled water before measurement.

### 4.6. Reverse Transcriptase Polymerase Chain Reaction (RT-PCR) of the Cultured Hepatocytes and the In Vivo Rat Model

Total RNA was extracted from liver samples or the cultured hepatocytes using Sepasol I Super G (Nacalai Tesque Inc., Kyoto, Japan), which contains a guanidinium thiocyanate–phenol–chloroform mixture [57]. For strand-specific RT-PCR, cDNA was first synthesized from total RNA with an oligo (dT) primer for mRNAs. Touchdown quantitative PCR was then performed using the primer pairs listed in Appendix A with Rotor-Gene Q (Qiagen, Valencia, CA, USA). The values of the mRNA levels obtained were normalized to those of elongation factor-1α (EF) mRNA. The normalized value of each gene in HIRI + PH for 3 h or IL-1β only for 4 h was set as 1.0 or 100%, respectively.

### 4.7. EMSA of the Nuclear Extracts

The EMSA was performed using a previously described method [49]. Nuclear extracts were prepared from the liver samples or the cultured hepatocytes. Protein concentrations of nuclear extracts were measured using the Bradford method. A double-stranded DNA probe containing a κB site (sense strand: 5′-AGTTGAGGGGACTTTCCCAGGC) was labeled with [γ-^32^P]-ATP (PerkinElmer Inc., Waltham, MA, USA). The nuclear extracts (4 μg of nuclear protein per lane) incubated with the labeled DNA probes (40,000 dpm) were resolved by polyacrylamide gel electrophoresis. Autoradiography was used to analyze the dried gels. The gel-shift band intensities were quantified using ImageJ 1.53k (National Institutes of Health).

### 4.8. Determination of Nitric Oxide Concentration and LDH Activity in the Cultured Hepatocytes

After the hepatocytes were incubated for 8 h with IL-1β and SFN, the medium was collected. Nitrite, which is a stable metabolite of nitric oxide, in the medium was measured in triplicate by the Griess method [47,58]. Cell viability was evaluated based on the activity of LDH released into the medium using a Cytotoxicity LDH Assay Kit-WST (code CK12, Dojindo Laboratories, Kumamoto, Japan).

### 4.9. Statistical Analyses

Quantitative results were obtained from three or four rats per group or from three independent experiments, and the means and standard deviations (SDs) were calculated. A one-way analysis of variance (ANOVA) followed by a Tukey–Kramer multiple comparisons test was performed to analyze differences. Survival curves were obtained using the Kaplan–Meier method and statistically analyzed using a log-rank test. Statistical analyses were performed using the JMP 16.2 Statistics software (SAS Institute Inc., Carly, NC, USA). *p*-values of <0.05 and *p* < 0.01 were considered significant versus the naïve rats, HIRI + PH-only rats, or IL-1β alone.

## 5. Conclusions

SFN suppressed liver injury after HIRI + PH administration, which may be due, in part, to the suppression of inflammatory cytokine and chemokine expression and the increased expression of anti-inflammatory cytokine and anti-apoptotic genes. The suppression of the expression of these inflammation-related genes may be associated with the inactivation of NF-κB in hepatocytes. Thus, SFN has powerful hepatoprotective effects against hepatic ischemia/reperfusion injury induced after occlusion of blood flow performed for partial hepatectomy. Further in-depth research is needed to investigate its clinical potential in the development of novel therapeutic agents.

## Figures and Tables

**Figure 1 ijms-25-00579-f001:**
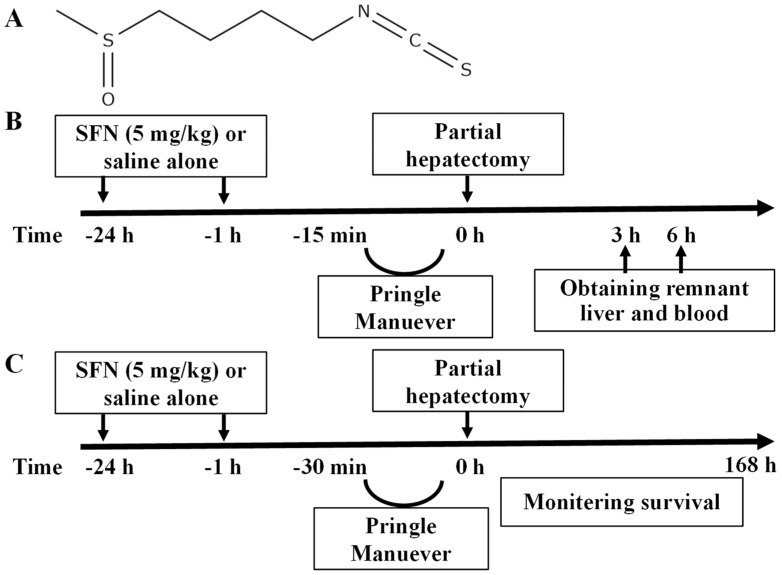
Study design. (**A**) Molecular structure of sulforaphane (SFN). In the hepatic ischemia/reperfusion injury and partial hepatectomy (HIRI + PH) group, the rats underwent 70% hepatectomy following HIRI after vehicle administration preoperatively. In the HIRI + PH and SFN groups, the rats were administered SFN (5 mg/kg) 1 and 24 h before HIRI + PH treatment. (**B**) Blood and liver samples were collected at the indicated time after HIRI + PH treatment. (**C**) Survival was monitored for 168 h after HIRI + PH treatment.

**Figure 2 ijms-25-00579-f002:**
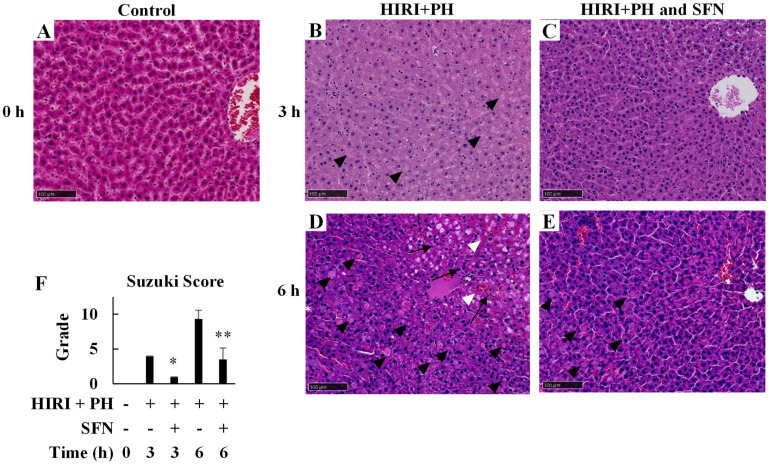
Effects of sulforaphane (SFN) on hepatic pathological changes in the livers of hepatic ischemia/reperfusion injury and partial hepatectomy (HIRI + PH)−treated rats. Representative liver histology of HIRI + PH treatment. Liver sections of (**A**) naïve rats and rats treated with (**B**,**D**) HIRI + PH and (**C**,**E**) HIRI + PH and SFN were stained with hematoxylin and eosin (H&E) (magnification ×200; bar = 100 μm). Note the areas of focal necrosis (arrows), inflammatory cell infiltration (white asterisk), massive hemorrhage (white arrowheads), and ballooning degeneration of the hepatocytes (black arrowheads) in HIRI + PH rats. (**F**) Suzuki histological grading in the livers. The values in the bar graphs represent the mean ± standard deviation (SD; *n* = 3–4 rats per time-point per group). * *p* < 0.05 and ** *p* < 0.01 versus HIRI + PH−only rats.

**Figure 3 ijms-25-00579-f003:**
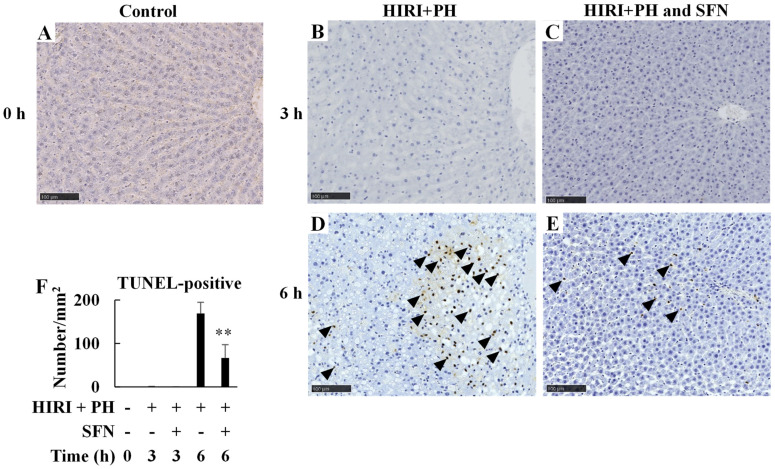
Effects of sulforaphane (SFN) on apoptosis in the livers of hepatic ischemia/reperfusion injury and partial hepatectomy (HIRI + PH)−treated rats. Representative liver histology of HIRI + PH treatment. Liver sections of (**A**) naïve rats and rats treated with (**B**,**D**) HIRI + PH and (**C**,**E**) HIRI + PH and SFN were stained with terminal deoxynucleotidyl transferase-mediated deoxyuridine nick-end labeling (TUNEL) (magnification ×200; bar = 100 μm). Note the brown nuclei in the positive cells (black arrowheads). (**F**) The numbers of TUNEL−positive cells per square millimeter were counted. The values in the bar graphs represent the mean ± standard deviation (SD; *n* = 3–4 rats per time-point per group). ** *p* < 0.01 versus HIRI + PH−only rats.

**Figure 4 ijms-25-00579-f004:**
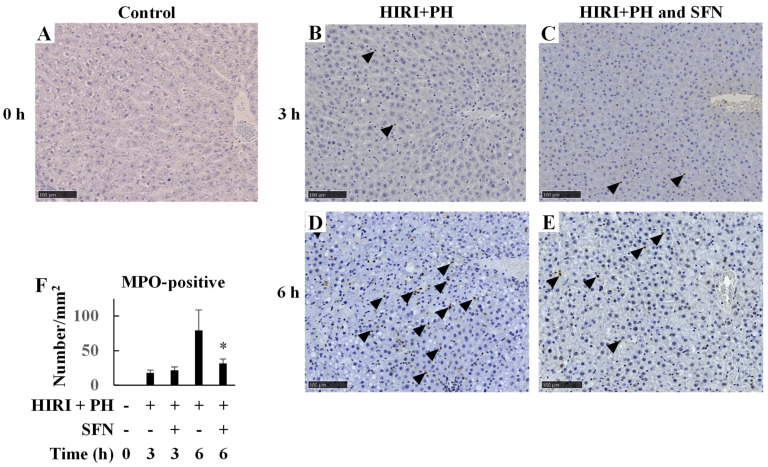
Effects of sulforaphane (SFN) on neutrophil infiltration in the livers of hepatic ischemia/reperfusion injury and partial hepatectomy (HIRI + PH)−treated rats. Representative liver histology of HIRI + PH treatment. The liver sections of (**A**) naïve rats and rats treated with (**B**,**D**) HIRI + PH and (**C**,**E**) HIRI + PH and SFN were stained with anti-myeloperoxidase (MPO) antibody (magnification ×200; bar = 100 μm). Note the brown staining in the positive cells (black arrowheads). (**F**) The numbers of MPO−positive cells per square millimeter were counted. The values in the bar graph represent the mean ± standard deviation (SD; *n* = 3–4 rats per time-point per group). * *p* < 0.05 versus HIRI + PH−only rats.

**Figure 5 ijms-25-00579-f005:**
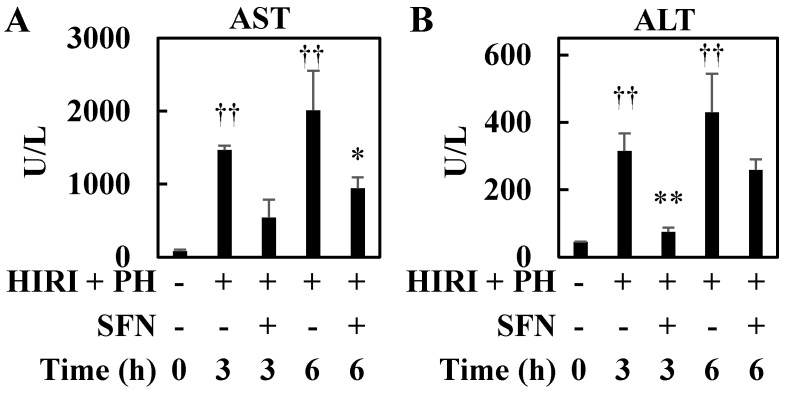
Effects of sulforaphane (SFN) on liver damage and functions in hepatic ischemia/reperfusion injury and partial hepatectomy (HIRI + PH)−treated rats. The serum levels of (**A**) aspartate aminotransferase (AST) and (**B**) alanine aminotransferase (ALT) were measured at the indicated time after HIRI + PH treatment. The values in the bar graphs represent the mean ± standard deviation (SD; *n* = 3–4 rats per time-point per group). †† *p* < 0.01 versus naïve rats. * *p* < 0.05 and ** *p* < 0.01 versus HIRI + PH−only rats.

**Figure 6 ijms-25-00579-f006:**
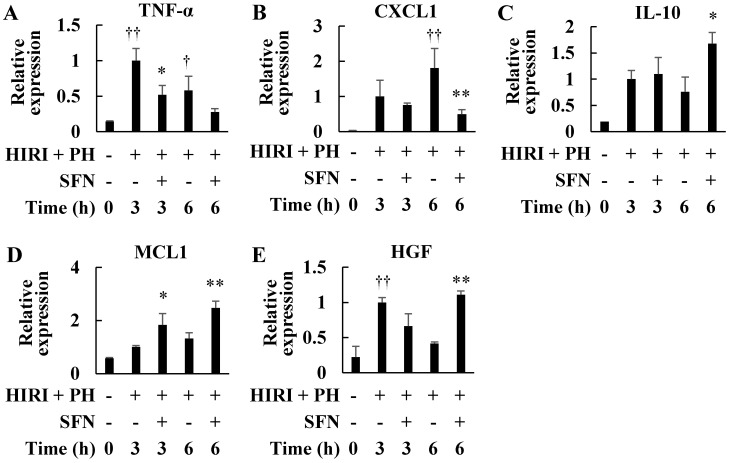
Effects of sulforaphane (SFN) on the mRNA expression of inflammatory and apoptosis- and liver regeneration-related genes in the livers of hepatic ischemia/reperfusion injury and partial hepatectomy (HIRI + PH)−treated rats. (**A**) Tumor necrosis factor-alpha (TNF−α), (**B**) C−X−C motif chemokine ligand 1 (CXCL1), (**C**) interleukin 10 (IL−10), (**D**) myeloid cell leukemia 1 (MCL1), and (**E**) hepatocyte growth factor (HGF) mRNA expression was analyzed by RT−PCR. The values in the bar graphs represent the mean ± standard deviation (SD; *n* = 3–4 rats per time−point per group). † *p* < 0.05 and †† *p* < 0.01 versus naïve rats. * *p* < 0.05 and ** *p* < 0.01 versus HIRI + PH−only rats.

**Figure 7 ijms-25-00579-f007:**
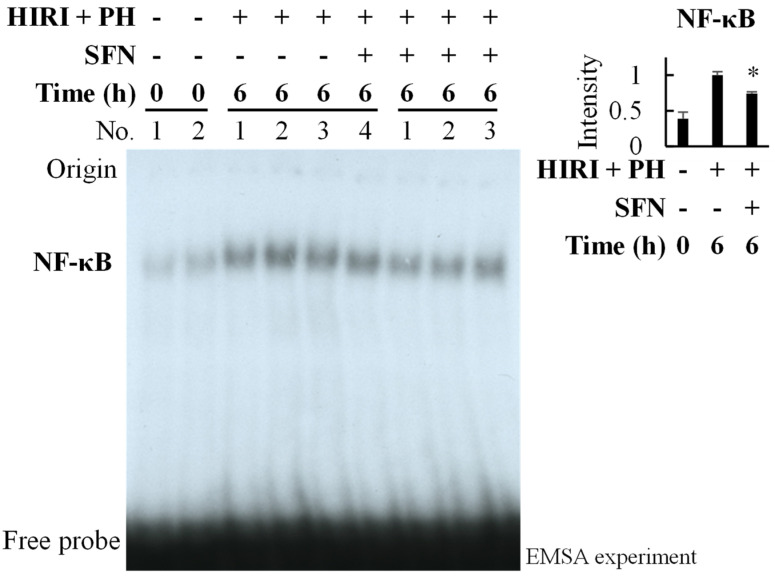
Effect of sulforaphane (SFN) treatment on nuclear factor-kappa B (NF−κB) activation in the livers of hepatic ischemia/reperfusion injury and partial hepatectomy (HIRI + PH)−treated rats. Binding capacities to the DNA probe harboring κB site in nuclear extracts of rat liver were analyzed by an electrophoretic mobility shift assay (EMSA). The mean intensity of the bands from rats treated with HIRI + PH only was set to 1.0 (**right**). * *p* < 0.05 versus HIRI + PH−only rats.

**Figure 8 ijms-25-00579-f008:**
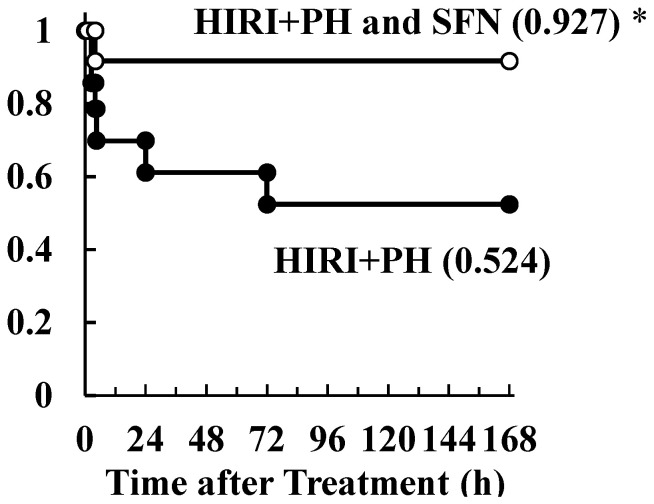
Effects of sulforaphane (SFN) on hepatic ischemia/reperfusion injury and partial hepatectomy (HIRI + PH)−treated rat survival with a longer ischemic duration. Kaplan−Meier survival curves of the cumulative survival of the following treatment groups: HIRI + PH−only (filled circles; *n* = 14) and HIRI + PH and SFN (open circles; *n* = 12). The values represent survival at 168 h after HIRI + PH treatment. * *p* < 0.05 versus HIRI + PH-only rats.

**Figure 9 ijms-25-00579-f009:**
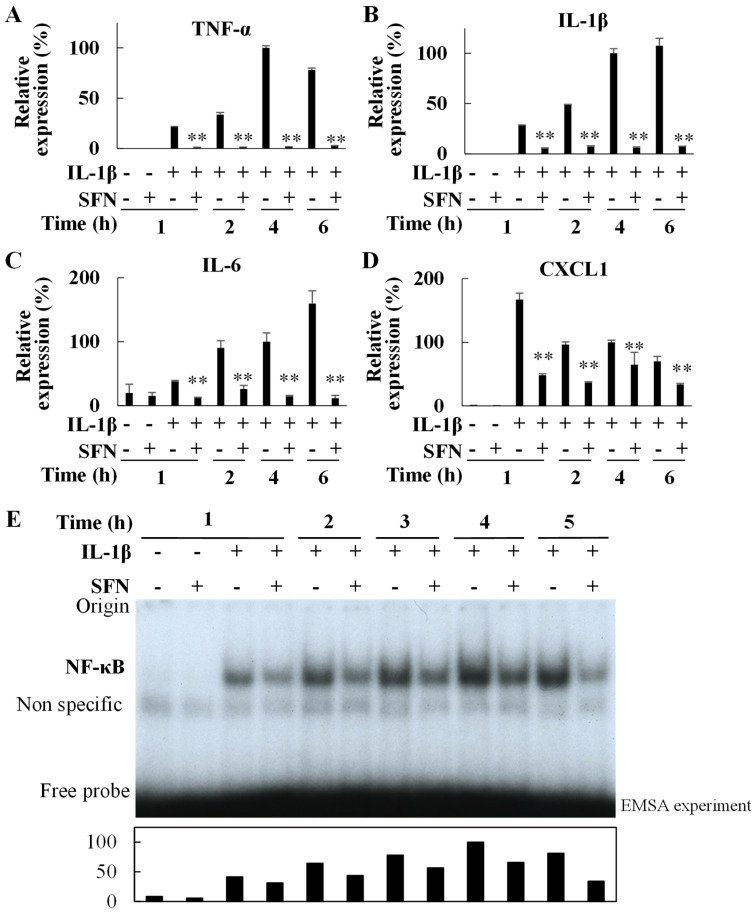
Effects of sulforaphane (SFN) on the inflammatory gene expression and nuclear factor-kappa B (NF-κB) activation in interleukin 1β (IL−1β)−stimulated hepatocytes. Cells were treated with IL-1β (1 nM) in the presence or absence of SFN (10 μM) for the indicated time. The effect of SFN treatment on the expression of (**A**) tumor necrosis factor-alpha (TNF−α), (**B**) IL−1β, (**C**) IL−6, (**D**) C−X−C motif chemokine ligand 1 (CXCL1) mRNA. The values in the bar graphs represent the mean ± standard deviation (SD; *n* = 3 experiments). The relative expression of each gene after 4 h of IL−1β (+) and SFN (−) treatment was set as 100%. ** *p* < 0.01 versus IL−1β alone. (**E**) The effect of SFN treatment on NF−κB activation analyzed by an electrophoretic mobility shift assay (EMSA). The arbitrary units of NF−κB activation were determined by setting the band intensity at 4 h of IL−1β (+) and SFN (−) treatment as 100 (lower).

## Data Availability

The data presented in this study are available upon request from the corresponding author.

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
