# Peer review of "Sulforaphane Is Protective against Warm Ischemia/Reperfusion Injury and Partial Hepatectomy in Rats"

_ijms, 2024, doi:10.3390/ijms25010579_

Round 1
Reviewer 1 Report
Comments and Suggestions for Authors
In this manuscript, several drawbacks exist, particularly in the experimental design and hypothesis, which affected the validity of the scientific conclusion of the study. Besides, wide revisions are needed as follows:
Major concerns:
1. The authors tested one dose of sulforaphane. Testing at least three doses was highly recommended to assess the underlying mechanisms accurately.
2. Why did the authors use two different strains of rats in the in vitro and in vivo study (i.e., Sprague-Dawley rats for the invivo study and Wistar rats for the in vitro study)?
3. A control group without HIRI+PH should be included in the study to give accurate comparisons.
4. Fig. 2. The histopathological and immunohistochemical figures are considered a major drawback of the study as they are of poor quality and low resolution. The authors should add arrows denoting the lesions and clarify the lesions in each group in the legend. Also, the positive reactions in the immunohistochemical figures are non-specific and unclear. Additionally, the authors have not clarified how many slides were made per animal. Did only one researcher carry out the analysis of these slides?
Other comments:
1. Lines 55-56: the references to animal studies are missed.
2. All figure legends and table footnotes should clarify the full terms of all abbreviations used. Clarify the number of replicates n=?
3. Results:
- The exact P-value must be provided in the result section to understand statistical significance clearly. Also, the authors should describe how much change was induced by SFN in the results compared with the HIRI+PH.
- The figures are of low quality and need to be reconstructed. Clarify that the data has been presented as means ± SD or SE.
4. Lines 174-190 should be transferred to the introduction.
5. Material and methods:
- Clarify the purity and molecular weight of SFN.
- Lines 248: clarify the initial weight of rats used.
- Complete information on kits used should be added as trademark, city, and country of origin, as well as detection range, sensitivity, and inter- and intra-assay.
- Statistical analysis:
• Line 320: specify three or four rats per group.
• Clarify that the data has been presented as means ± SD or SE.
• Does data meet the assumption of homogeneity of variances and normal distribution? Clarify if the authors run a homogeneity or normality test.
Comments on the Quality of English LanguageGood language
Author Response
First of all, we thank the reviewer for their invaluable comments and suggestions. We have addressed all the comments and revised the text. Please find our responses to the comments below.
In this manuscript, several drawbacks exist, particularly in the experimental design and hypothesis, which affected the validity of the scientific conclusion of the study. Besides, wide revisions are needed as follows:
Major concerns:
- The authors tested one dose of sulforaphane. Testing at least three doses was highly recommended to assess the underlying mechanisms accurately.
-In the in vitro model of liver injury, it has been established that drug concentrations are determined based on half-maximal inhibitory concentration (IC50) values, which are concentrations that inhibit the induction of nitric oxide (NO) production by 50%, and that these drug concentrations are used to assess the mechanisms underlying the anti-inflammatory effects of the drugs in primary cultured rat hepatocytes (Takimoto et al.; Fujii et al.; Yamauchi et al.). First, we investigated whether SFN inhibits the production of nitric oxide (NO), one of the most typical inflammatory factors, using the medium containing four doses of SFN (1 to 10 μM) in IL-1β-stimulated rat primary hepatocytes. SFN dose-dependently suppressed NO production without showing cytotoxicity up to 10 μM. The IC50 value of SFN was calculated to be 1.20 μM. The suppression of NO production is closely related to the suppression of the mRNA expression of the inflammatory genes by not only a variety of drugs, but also by several natural phytochemicals (Ozaki et al.; Kotsuka et al.; Ningsih et al.; Nakatake et al.). Taken together, we determined the concentration of SFN at which to assess the underlying mechanisms to be 10 µM.
We described these processes for determining concentrations in the manuscript, citing the studies that showed a link between suppression of NO production and suppression of gene expression. We also added the method of measuring NO production to the Materials and methods, the results of the dose-dependent effects on NO production under the inflammatory stimuli of SFN to the Supplementary figure, and the references below.
Lines 188-195, 361-364
Takimoto, Y., Qian, H. Y., Yoshigai, E., Okumura, T., Ikeya, Y., & Nishizawa, M. (2013). Gomisin N in the herbal drug gomishi (Schisandra chinensis) suppresses inducible nitric oxide synthase gene via C/EBPβ and NF-κB in rat hepatocytes. Nitric oxide, 28, 47–56. https://doi.org/10.1016/j.niox.2012.10.003
Fujii, A., Okuyama, T., Wakame, K. et al. Identification of anti-inflammatory constituents in Phellodendri Cortex and Coptidis Rhizoma by monitoring the suppression of nitric oxide production. J Nat Med 71, 745–756 (2017). https://doi.org/10.1007
Ozaki, H.; Nishidono, Y.; Fujii, A.; Okuyama, T.; Nakamura, K.; Maesako, T.; Shirako, S.; Nakatake, R.; Tanaka, K.; Ikeya, Y.; et al. Identification of Anti-Inflammatory Compounds from Peucedanum praeruptorum Roots by Using Nitric Oxide-Producing Rat Hepatocytes Stimulated by Interleukin 1β. Molecules 2023, 28, 5076. https://doi.org/10.3390/molecules28135076
Kotsuka, M., Hashimoto, Y., Nakatake, R., Okuyama, T., Hatta, M., Yoshida, T., Okumura, T., Nishizawa, M., Kaibori, M., & Sekimoto, M. (2022). Omeprazole Increases Survival Through the Inhibition of Inflammatory Mediaters in Two Rat Sepsis Models. Shock, 57(3), 444–456. https://doi.org/10.1097/SHK.0000000000001897
Ningsih, F. N., Okuyama, T., To, S., Nishidono, Y., Okumura, T., Tanaka, K., Ikeya, Y., & Nishizawa, M. (2020). Comparative Analysis of Anti-inflammatory Activity of the Constituents of the Rhizome of Cnidium officinale Using Rat Hepatocytes. Biological & pharmaceutical bulletin, 43(12), 1867–1875. https://doi.org/10.1248/bpb.b20-00416
Nakatake, R.; Hishikawa, H.; Kotsuka, M.; Ishizaki, M.; Matsui, K.; Nishizawa, M.; Yoshizawa, K.; Kaibori, M.; Okumura, T. The Proton Pump Inhibitor Lansoprazole Has Hepatoprotective Effects in In Vitro and In Vivo Rat Models of Acute Liver Injury. Dig Dis Sci 2019, 64, 2854-2866, DOI:10.1007/s10620-019-05622-6.
- Why did the authors use two different strains of rats in the in vitro and in vivo study (i.e., Sprague-Dawley rats for the in vivo study and Wistar rats for the in vitro study)?
-The in vivo rat model of HIRI has been established in male Sprague-Dawley rats (Ishizaki et al.). In addition, Sprague-Dawley rats were used in many papers, as indicated in the systematic review (Nakatake et al.). On the other hand, the in vitro model of liver injury has been established in male Wistar rats (Tanaka et al.; Matsui et al.; and so on). Therefore, the respective strains were applied to each in vivo or in vitro model in this study. We added this explanation to the Materials and Methods.
Line 188-195
Ishizaki, M.; Kaibori, M.; Uchida, Y.; Tanaka, H.; Ozaki, T.; Saito, T.; Matsui, K.; Kamiyama, Y.; Nishizawa, M.; Okumura, T. Protective effect of FR183998, a Na+/H+ exchanger inhibitor, and its inhibition of inducible nitric oxide synthase induction in hepatic ischemia-reperfusion injury in rats. Am J Transplant 2008, 8, 496-496.
Nakatake, R.; Schulz, M.; Kalvelage, C.; Benstoem, C.; Tolba, R.H. Effects of iNOS in Hepatic Warm Ischaemia and Reperfusion Models in Mice and Rats: A Systematic Review and Meta-Analysis. Int J Mol Sci. 2022, 23
Tanaka, H.; Uchida, Y.; Kaibori, M.; Hijikawa, T.; Ishizaki, M.; Yamada, M.; Matsui, K.; Ozaki, T.; Tokuhara, K.; Kamiyama, Y.; et al. Na+/H+ exchanger inhibitor, FR183998, has protective effect in lethal acute liver failure and prevents iNOS induction in rats. J Hepatol 2008, 48, 289-299, DOI:10.1016/j.jhep.2007.09.017.
Matsui, K.; Kawaguchi, Y.; Ozaki, T.; Tokuhara, K.; Tanaka, H.; Kaibori, M.; Matsui, Y.; Kamiyama, Y.; Wakame, K.; Miura, T.; et al. Effect of active hexose correlated compound on the production of nitric oxide in hepatocytes. J Parenter Enteral Nutr 2007, 31, 373-380, DOI:10.1177/0148607107031005373.
- A control group without HIRI+PH should be included in the study to give accurate comparisons.
-According to the reviewer's comments, we added the images of the control group without HIRI+PH (naïve rats) to Figure 2-4.
- Fig. 2. The histopathological and immunohistochemical figures are considered a major drawback of the study as they are of poor quality and low resolution. The authors should add arrows denoting the lesions and clarify the lesions in each group in the legend. Also, the positive reactions in the immunohistochemical figures are non-specific and unclear. Additionally, the authors have not clarified how many slides were made per animal. Did only one researcher carry out the analysis of these slides?
-According to the reviewer's comments, we improved the quality and resolution of the images and employed higher magnification. In addition, we added annotations clarifying pathological changes, including areas of focal necrosis, inflammatory cell infiltration, massive hemorrhage, and ballooning degeneration of the hepatocytes or positive staining to the image. To allow for as large an image as possible, Figure 2 was divided and the image was reconstructed. Three to four animals per group and three to five sites per animal were used for slide preparation. We added these description and details on the method of pathology evaluation and the analyst to the Materials and Methods.
Pages 3-5
Other comments:
- Lines 55-56: the references to animal studies are missed.
-As mentioned below, we revised the text to cite the references because the description of animal experiments was transferred from the discussion to the introduction.
Lines 54-56, 58-63
- All figure legends and table footnotes should clarify the full terms of all abbreviations used. Clarify the number of replicates n=?
-We added the full terms of all abbreviations in all figure legends and table footnotes. We also added the number of rats per group for sample or survival experiments to Materials Methods and the number of replicates for the RT-PCR experiment to figure legend of Figure 9 (old Figure 7).
Lines 96-104, 111-118, 125-131, 139-143, 17-162, 170-174, 181-182, 201-209,
- Results:
- The exact P-value must be provided in the result section to understand statistical significance clearly. Also, the authors should describe how much change was induced by SFN in the results compared with the HIRI+PH.
-We added each P value and the change due to SFN.
Lines 93, 108, 122, 136-137, 147-149, 151, 153-155, 167-168, 196-197
- The figures are of low quality and need to be reconstructed. Clarify that the data has been presented as means ± SD or SE.
-As mentioned above, the figures have been replaced with high-resolution ones. We added to the figure legends that the values in the bar graphs represent the mean ± standard deviation (SD). We specified that the means ± standard deviations (SD) were calculated in the Materials and Methods.
Line 370
- Lines 174-190 should be transferred to the introduction.
-We transferred the description in lines 174-190 to the introduction and revised the text.
Lines 59-72
- Material and methods:
- Clarify the purity and molecular weight of SFN.
-We added the purities and molecular weight of SFN in subsection 4.1.
Lines 268-269
- Lines 248: clarify the initial weight of rats used.
-We added the initial weight of rats used to subsection 4.2. Since rat weight at the time of purchase was incorrectly stated in subsection 4.2, we revised the text in subsection 4.1.
Lines 284, 271
- Complete information on kits used should be added as trademark, city, and country of origin, as well as detection range, sensitivity, and inter- and intra-assay.
-We added complete information on kits to Material and methods.
Lines 332-337
- Statistical analysis:
- Line 320: specify three or four rats per group.
-We revised the text.
Line 369
- Clarify that the data has been presented as means ± SD or SE.
-As mentioned above, we clarified the data has been presented as means ± SD.
Line 370
- Does data meet the assumption of homogeneity of variances and normal distribution? Clarify if the authors run a homogeneity or normality test.
-The point made by the reviewer was correct, a homogeneity or normality test was necessary before performing ANOVA and a statistical test, but it was difficult to apply these tests because the data were obtained by analysis with a minimum number of animals according to the guidelines for animal experiments. We have published many studies using primary cultured hepatocytes, as cited in the text, and in those studies, we have performed statistical analysis in parametric studies.
We consulted these points regarding statistics with a professor in the Department of Mathematics at Kansai Medical University, who is an expert in statistics. Appreciation for the mathematics professor was added to the Acknowledgment.
Reviewer 2 Report
Comments and Suggestions for Authors
The study by Nakatake et al. investigates the impact of Sulforaphane on inflammatory mediators induced by warm hepatic ischemia/reperfusion injury. Rats underwent hepatic ischemia followed by partial hepatectomy, and SFN was injected intraperitoneally before ischemia. SFN demonstrated hepatoprotective effects by reducing liver enzyme activities, pathological liver injury, and mortality. SFN also suppressed inflammatory cytokines and nuclear factor-kappa B activation in hepatocytes, suggesting its potential to inhibit the induction of inflammatory mediators in HIRI.
1. Prior studies (PMID: 34320386, 25924817, 25788192) have explored the impact of sulforaphane on HIRI, uncovering certain mechanisms. However, this present study does not contribute novel insights to the existing knowledge. While the inclusion of partial hepatectomy with HIRI is commendable, the author misses an opportunity to leverage the hepatectomy model to analyze the influence of sulforaphane on liver regeneration. This addition could have brought a new dimension to the study.
2. NFkB serves as a typical downstream factor in the inflammatory response. Consequently, the inhibition of NFkB is expected if sulforaphane possesses anti-inflammatory properties. To discern the precise impact of sulforaphane on inflammation, it is imperative to delve into upstream signaling pathways for a comprehensive understanding.
3. The histological images in this study are inadequately presented. The author should specify the magnification used and consider employing higher magnification to better illustrate histological changes. It is recommended to enhance clarity by incorporating arrows to indicate each observed change, such as necrosis and inflammation.
4. The TUNEL staining in Figure 2H appears nonspecific; it resembles a necrotic area rather than demonstrating positive TUNEL staining in the nucleus.
5. In in vivo models, it is advisable to measure inflammatory cytokines such as IL-6 and IL-1β. Assessing the levels of IL-6, in particular, can provide valuable insights into the regeneration of remnant livers.
6. The quality of WB image needs to be improved, and the internal control is missing.
Comments on the Quality of English Languageminor editing
Author Response
ijms-2774544, entitled “Sulforaphane Is Protective Against Warm Ischemia/Reperfusion Injury and Partial Hepatectomy in Rats”
First of all, we thank the reviewer for their invaluable comments and suggestions. We have addressed all the comments and revised the text. Please find our responses to the comments below.
The study by Nakatake et al. investigates the impact of Sulforaphane on inflammatory mediators induced by warm hepatic ischemia/reperfusion injury. Rats underwent hepatic ischemia followed by partial hepatectomy, and SFN was injected intraperitoneally before ischemia. SFN demonstrated hepatoprotective effects by reducing liver enzyme activities, pathological liver injury, and mortality. SFN also suppressed inflammatory cytokines and nuclear factor kappa B activation in hepatocytes, suggesting its potential to inhibit the induction of inflammatory mediators in HIRI.
- Prior studies (PMID: 34320386, 25924817, 25788192) have explored the impact of sulforaphane on HIRI,
uncovering certain mechanisms. However, this present study does not contribute novel insights to the existing knowledge. While the inclusion of partial hepatectomy with HIRI is commendable, the author misses an opportunity to leverage the hepatectomy model to analyze the influence of sulforaphane on liver regeneration. This addition could have brought a new dimension to the study.
-According to the Reviewer’s suggestion, we examined the effects of SFN on mRNA expression of hepatocyte growth factor (HGF) and added the results to the Results and the Discussion. We measured the weight of remnant liver lobes 6 h after 15 min of ischemia and partial hepatectomy and found no difference between the HIRI+PH only group and the HIRI+PH and SFN group. Further investigation of the effect of SFN on liver regeneration by measuring the weight of rat livers after 168 h in survival experiments was partially unsuccessful as approximately half of the rats died due to the lethal conditions induced by the prolonged ischemic time, and it was not clear whether SFN would affect the promotion of liver regeneration. In addition, we cited a previous study in which activation of the transcription factor Nrf2, an intracellular target of SFN, promotes liver regeneration (Chan et al.) and mentioned liver regeneration in HIRI+PH treatment in the Discussion.
Lines 153-155, 252-258
Chan, B.K.Y.; Elmasry, M.; Forootan, S.S.; Russomanno, G.; Bunday, T.M.; Zhang, F.; Brillant, N.; Starkey Lewis, P.J.; Aird, R.; Ricci, E.; Andrews, T.D., et al. Pharmacological Activation of Nrf2 Enhances Functional Liver Regeneration. Hepatology 2021, 74, 973-986. DOI:10.1002/hep.31859. (PMID: 33872408)
- NFkB serves as a typical downstream factor in the inflammatory response. Consequently, the inhibition of NFkB is expected if sulforaphane possesses anti-inflammatory properties. To discern the precise impact of sulforaphane on inflammation, it is imperative to delve into upstream signaling pathways for a comprehensive understanding.
-We appreciated the Reviewer’s comments. In IL-1β-stimulated primary cultured rat hepatocytes, SNF did not affect IκB or Akt, which are involved in NF-κB activation; elucidating the mechanism by which SFN suppresses NF-κB activity is a future challenge. These were added to the Discussion.
Lines 247-251
The histological images in this study are inadequately presented. The author should specify the magnification used and consider employing higher magnification to better illustrate histological changes. It is recommended to enhance clarity by incorporating arrows to indicate each observed change, such as necrosis and inflammation.
-As mentioned above, we improved the quality and resolution of the images and employed higher magnification. To allow for as large an image as possible, Figure 2 was divided and the image was reconstructed. In addition, we added annotations clarifying pathological changes, including areas of focal necrosis, inflammatory cell infiltration, massive hemorrhage, and ballooning degeneration of the hepatocytes or positive staining to the image.
Pages 3-5
- The TUNEL staining in Figure 2H appears nonspecific; it resembles a necrotic area rather than demonstrating positive TUNEL staining in the nucleus.
-As mentioned above, the quality and resolution of the images were improved. In addition, annotations indicating positive staining were added to the images.
- In in vivo models, it is advisable to measure inflammatory cytokines such as IL-6 and IL-1β. Assessing the levels of IL-6, in particular, can provide valuable insights into the regeneration of remnant livers.
-The reviewer's advice is invaluable, and we have examined the effect of SFN on the expression of IL-6 and IL-1β in the liver of HIRI+PH-treated rats. Their expression levels were lower than those induced in primary cultured hepatocytes and the changes were also so small that they were not statistically analyzed and not shown in the figures. The discussion about liver regeneration is mentioned above.
- The quality of WB image needs to be improved, and the internal control is missing.
-Figure 7 and 9E (old Figures 5 and 7E) are the results by EMSA experiments, not by WB experiments, so there are no internal controls. The nuclear extracts were adjusted to the equivalent of 4 µg protein per lane. This was added to the Materials and Methods. We also noted the EMSA experiment in Figures and Figure legends. We replaced the images with higher-quality ones.
Lines 208-209, 355-359
Round 2
Reviewer 1 Report
Comments and Suggestions for Authors
The authors properly addressed the reviewer's comments and performed the required modifications.
Author Response
We thank the reviewer for taking the time to review this manuscript.
The authors properly addressed the reviewer's comments and performed the required modifications.
-We appreciate the reviewer's constructive review of our manuscript.
Reviewer 2 Report
Comments and Suggestions for Authors
The authors have answered most of my questions. However, the study of liver regeneration is questionable, The prime phase for liver regeneration for rats is 24 h after PH. Measuring liver weight alone is not adequate to express liver regeneration. Liver to body weight ratio should be used. In addition, IHC staining like Ki-67, PCNA, PH3 are suggested to evaluate liver regeneration.
Author Response
We thank the reviewer for taking the time to review this manuscript. We have addressed all the comments and revised the text. Please find our responses to the comments below and the corresponding revisions/corrections in red in the re-submitted files.
The authors have answered most of my questions. However, the study of liver regeneration is questionable, The prime phase for liver regeneration for rats is 24 h after PH. Measuring liver weight alone is not adequate to express liver regeneration. Liver to body weight ratio should be used. In addition, IHC staining like Ki-67, PCNA, PH3 are suggested to evaluate liver regeneration.
-We agree that the ratio of liver to body weight is adequate to express liver regeneration and would like to correct our previous response; the liver weights were measured in the HIRI+PH-treated rats, and the ratio was calculated. The process of liver regeneration in rats begins with an increase in the levels of various signaling molecules 3 h after hepatectomy, peaks at two to three days, and ends with the enlargement of the remnant lobe to the original liver size at five to seven days (Wang et al.). In this study, we did not find any significant difference in liver to body weight ratio after 6 h (15 min of the Pringle maneuver) and 168 h (lethal 30 min of the Pringle maneuver). We considered it less possible to assess the effect on liver regeneration in 48-72 h after HIRI+PH treatment in the case of milder injuries due to 15 min of PM. As pointed out, these experiments would bring a new dimension to the research, but it will be difficult to obtain results within 10 days to respond to peer review. We revised the text in the discussion and added the new citations below.
Round 3
Reviewer 2 Report
Comments and Suggestions for Authors
My questions have been addressed.